# Design and Implementation of a Framework for Smart Home Automation Based on Cellular IoT, MQTT, and Serverless Functions

**DOI:** 10.3390/s23094459

**Published:** 2023-05-03

**Authors:** Marco Esposito, Alberto Belli, Lorenzo Palma, Paola Pierleoni

**Affiliations:** Department of Information Engineering (DII), Università Politecnica delle Marche, 60131 Ancona, Italy

**Keywords:** internet of things, home automation, smart appliance, narrowband IoT, MQTT, serverless

## Abstract

Smart objects and home automation tools are becoming increasingly popular, and the number of smart devices that each dedicated application has to manage is increasing accordingly. The emergence of technologies such as serverless computing and dedicated machine-to-machine communication protocols represents a valuable opportunity to facilitate management of smart objects and replicability of new solutions. The aim of this paper is to propose a framework for home automation applications that can be applied to control and monitor any appliance or object in a smart home environment. The proposed framework makes use of a dedicated messages-exchange protocol based on MQTT and cloud-deployed serverless functions. Furthermore, a vocal command interface is implemented to let users control the smart object with vocal interactions, greatly increasing the accessibility and intuitiveness of the proposed solution. A smart object, namely a smart kitchen fan extractor system, was developed, prototyped, and tested to illustrate the viability of the proposed solution. The smart object is equipped with a narrowband IoT (NB-IoT) module to send and receive commands to and from the cloud. In order to evaluate the performance of the proposed solution, the suitability of NB-IoT for the transmission of MQTT messages was evaluated. The results show how NB-IoT has an acceptable latency performance despite some minimal packet loss.

## 1. Introduction

Completely automated smart homes are on the way to becoming a well-established reality. The ever increasing number of smart objects in the smart home environment requires the definition of standardized and flexible protocols for commands and state information exchange, besides optimal strategies for processing a large number of commands. New technologies such as serverless computing and ad hoc communication protocols can be leveraged to manage a large fleet of smart objects, while also ensuring good accessibility and intuitive user interfaces. Serverless computing is an emerging cloud-computing paradigm that represents a promising development for many application fields, from micro- and web-services [1], to the the internet of things (IoT) [2,3], edge computing [4] and artificial intelligence (AI) [5], or even for domain-specific scientific applications [6,7]. Serverless computing operates at the crossing point of different models that have emerged alongside cloud computing, such as function as a service (FaaS) and pay-per-use models, and technologies such as event-driven programming, virtualization, and containerization [4]. Serverless computing comes with automated scalability and resource provisioning, meaning that resources assigned by the serverless platform scale depending on the number of functions that are running and the resource demand of each of those functions. These characteristics make serverless particularly appealing for applications that need to manage a large pool of users or, in the field of the internet of things and smart objects, a large number of devices [2].

The pioneering commercial platform for serverless computing was Lambda [8], developed by Amazon Web Services (AWS), the Amazon cloud platform. Lambda remains one of the most popular platforms for the development of serverless applications. A recent work by the authors of this paper [9] showed how AWS satisfies many key performance indicators in IoT applications, also compared to other competitors such as Microsoft Azure and Google cloud Platform, both of which now also offer serverless and FaaS services. Lambda also comes with extensive integration with Amazon Alexa [10], the voice assistant on Amazon Echo. Voice interfaces are seeing a wider use in different application fields, from voice assistants to home automation, as they constitute a more natural and intuitive interface to interact with smart objects and environments, as well as improving accessibility [11].

This paper proposes and experimentally evaluates a framework that represents a synergistic union of different cloud-computing and IoT-related technologies that can enable accessible and automated home environments. The framework is easy to replicate, is scalable, and is transparent to the underlying smart object or appliance that uses it due to the use of serverless technologies and a custom-defined command exchange protocol based on the message queue telemetry transport (MQTT) protocol. MQTT [12] is the main middleware service used in most cloud platforms for the IoT, therefore simplifying the interfacing of IoT devices and smart objects with the cloud. Compared to other protocols such as HTTP, which is also employed in many serverless solutions [3], MQTT messages have a small overhead that makes the protocol suitable for machine-to-machine applications. Furthermore, MQTT does not impose a format for its payload, making it extremely flexible [13], and the use of topics adds a further degree of freedom for the implementation of custom applications.

As the introduction of new communication technologies in such applications requires careful evaluation in terms of the overall application latency, the aim of this paper is to evaluate the performance of a framework for smart home appliances equipped with narrowband IoT (NB-IoT) modules to send and receive messages to/from the cloud. NB-IoT is a low-power wide area network (LPWAN) protocol by the Third-Generation Partnership Project (3GPP), designed to provide wide connectivity to a huge number of IoT devices in massive deployment scenarios and in scenarios that require good radio coverage and lower power consumption, such as smart metering [14,15]. NB-IoT devices can reach nominal life times of around 10 years with small and infrequent messaging [16], with an overall good power consumption performance compared to other communication technologies, both cellular and not [17]. Cellular technologies such as NB-IoT can be employed to embed additional services that require internet connectivity in home and kitchen appliances, adding negligible power consumption to their normal mode of operation and advancing the machines as a service (MaaS) model [18,19] and servitization of the products [20].

The framework proposed in this paper includes a smart IoT device making use of the aforementioned MQTT-based command protocol, and a serverless application on the cloud to manage devices and commands received through Amazon Alexa or through a dedicated application. The NB-IoT protocol was tested to determine if it is suitable to dispatch MQTT messages between MQTT clients. Automatic speech recognition (ASR) software already adds a delay to the overall application latency, and the execution of Lambda functions also adds a (minimal) latency, but the message exchange delay might severely degrade the user experience if it hinders the prompt execution of commands. For the demonstration of the proposed framework viability, a smart extractor fan system was designed, prototyped, and tested as a practical use case. An experimental test-bed with a NB-IoT node sending and receiving MQTT messages was therefore set up to evaluate the latency performance of the protocol in this context.

This work is organized as follows. Section 2 explains the main technologies employed in the proposed system, and illustrates some recent works in the field of home automation and vocal interfaces for smart homes. Section 3 goes into detail about the proposed system and the NB-IoT evaluation test-bed, while Section 5 gives the results of the evaluation. Lastly, Section 6 draws conclusions about the work and contributions of this paper.

## 2. Background

Many smart home automation papers propose APIs or practical implementations of smart device use cases, employing protocols ranging from HTTP to MQTT to CoAP, which has also shown potential for home automation applications [21]. Crisan et al. [22] proposed and developed an API for smart home automation called qtoggle to control any IoT device in the smart home environment with a TCP/IP stack via HTTP requests, regardless of the type of smart object or device. Sarkar et al. [23] proposed a framework to manage IoT devices in smart buildings using the serverless paradigm. Their proposed system consists of gas and humidity sensors and smart bulbs, and the serverless platform of choice was the OpenFaas platform. Froiz-Míguez et al. [24] proposed a framework for fog computing in smart homes called Ziwi, which enables the coexistence of Zigbee and Wifi devices on the same smart home framework. MQTT is used for its small code footprint and for addressing compatibility between smart devices. The increasing usage of MQTT in machine-to-machine applications has been stressed by Biswajeeban et al. [25], stemming from its lightweight design and its publish/subscribe architecture. Most frameworks employing this protocol require the implementation of security mechanisms (such as TLS) that add some extra overhead to the communication. As research on the topic of security in MQTT is increasing [26,27], various recent papers have also focused on the implementation of new security mechanisms designed specifically for MQTT-based smart home environments [28,29].

Different recent works explored the potentiality of integrating Amazon Alexa or, more generally, voice services into the smart environment to enable interactive and accessible interfaces. Iliev et al. [11] proposes a distributed framework for smart homes that uses natural language processing (NLP) to develop a human/smart environment interface. Their solution introduces a fog node for the deployment of intelligent user interface (IUI) software to reduce the voice interface response delay and to decrease bandwidth load. Krishnamurthi et al. [30] analyzed different data fusion and techniques for work offloading in IoT applications, such as fog and edge computing, which can find use in different IoT fields, including smart homes. The interface is designed to understand domain-specific natural language to support under-resourced languages that lack available ASR software, such as Amazon Alexa. Lee et al. [31] explored the possibility of using Amazon Alexa to build smart home services for the elderly or to enhance the effectiveness of existing services such as “carebots”. Bodgan et al. [32] proposed a framework that integrates voice assistants into smart offices for various tasks such as ambient control and interaction with various services.

This paper proposes a very flexible framework for smart home automation that embeds a voice interface, while in addition evaluating the use of cellular technologies for the dispatching of MQTT messages for commands and status updates. The introduction of cellular communication in these frameworks, if carefully evaluated, can aid in the implementation of different services in home and kitchen appliances, enabling, for example, predictive maintenance and remote monitoring services to be built on top of said frameworks. The flexibility of the serverless model and the use of MQTT, which is the main middleware in the cloud technologies described above, allowed for the integration of the different services provided by AWS, including voice interaction services. Furthermore, NB-IoT has been evaluated to test its efficiency in serving a framework based on MQTT. The remaining part of this section describes the technologies used for the design, testing, and evaluation of the proposed framework, embedding a smart extractor fan prototype. The details of the implementation and fusion of the aforementioned technologies are detailed in Section 3.

### 2.1. Narrowband IoT

NB-IoT is a cellular technology based on long-term evolution (LTE) introduced by 3GPP. NB-IoT reduces the LTE protocol stack functionalities to make it suitable for IoT applications. A brief comparison of NB-IoT with other LPWAN communication protocols aimed at comparable applications is in Table 1. Compared to protocols such as LoRaWAN and Sigfox, NB-IoT works in LTE bands and does not require the use of a gateway. It provides extended coverage to delay-tolerant, low-cost, low-energy IoT applications, reaching good energy performance even in the worse-case, continuous transmission scenarios when compared to other communication protocols [17]. It represents a valuable choice for application fields that require massive deployments, good coverage and low data rates, including smart metering [33], smart grids [34], smart cities [35], healthcare [36,37,38], and industrial applications [39]. To the best of the authors’ knowledge, it has still been seldom applied in home automation [40], mostly in home surveillance or metering applications [41,42], and rarely using MQTT. Cellular connectivity provides opportunities for enterprises to embed additional services into home appliances [20]. Furthermore, NB-IoT will be used in the machine-to-machine communication use case in 5G networks and will coexist with other services in the 5G environment [43]. While some works exist on the application layer performance over NB-IoT [44,45], this work mainly focuses on latency issues for the smart home use case, with the goal of ensuring a good user experience for the vocal interaction subsystem.

### 2.2. MQTT

MQTT is an OASIS standard messaging protocol designed to be lightweight and to have a small code footprint [12]. MQTT is one the main protocols that is used for the IoT and for the implementation of middleware services, alongside COAP, AMQP and HTTP [49]. It is nowadays used in many application scenarios, such as smart homes and smart buildings [50,51], smart cities [52], smart grids and energy management [53,54], smart agriculture [55], environmental monitoring and early warning [56], and so on. Furthermore, the main cloud platforms for the IoT use MQTT as their middleware service and brokering protocol [9].

MQTT is a publish/subscribe message protocol that decouples data producers and data consumers through the use of a central entity called “broker”, a server responsible for dispatching messages among MQTT clients. brokers contain a list of topics, which are strings that are organized hierarchically. The MQTT client can either subscribe to a topic, publish a message on a topic, or both. When a client publishes a message on a certain topic, the broker will forward it to every client that has subscribed to that topic. MQTT includes a series of security mechanisms, including password and username authentication, and authentication with transport layer certificate exchange. Data encryption can be guaranteed with protocols such as the transport layer security protocol (TLS), as is performed in AWS, or it can be implemented at the application layer to provide end-to-end encryption.

### 2.3. Amazon Web Services

Amazon Web Services is one of the most famous and oldest cloud platforms, offering services ranging from databases and data analytics to internet of things and mobile computing, besides many services dedicated to enterprises [57]. The smart appliance system proposed in this paper makes use of AWS IoT core, AWS Lambda, and two data storage services, AWS S3 and DynamoDB. Furthermore, Amazon Cloudwatch, a service used to monitor AWS resources and applications, is used to keep track of the performance and cost of each deployed service.

#### 2.3.1. AWS IoT Core

AWS IoT core is the AWS platform for the internet of things. It allows devices to connect to the cloud and other AWS services. The architecture of AWS IoT core is described in Figure 1. The central component of the architecture is the message broker, which dispatches messages to and from things (i.e., IoT devices) and other AWS services and user applications. AWS IoT core supports MQTT and MQTT over Websockets, but clients can also publish messages on the broker using HTTP. A rules engine is available to automatically process messages and to generate triggers. Each IoT device has an associated “shadow” to keep track of its status, and shadows are stored in a corresponding registry. Various security and identity mechanisms are employed to guarantee secure communication between the broker and devices, including TLS data encryption [58] and secure authentication with X.509 certificates [59].

#### 2.3.2. AWS Lambda

AWS Lambda is a serverless computing platform that lets users deploy applications and run code without managing servers, delegating to Lambda all the server and operating system maintenance, including capacity provisioning and automatic scaling of resources, while leaving the code as the sole responsibility of the developer. Lambda can be used to build IoT backends and countless other applications. Lambda functions are event-driven, meaning that they are instantiated to manage events. Each function is usually linked to an event source that generates triggers and will contain a series of handlers to manage those triggers. Various AWS services are event sources and can be configured to trigger Lambda functions, including Amazon Alexa skills and AWS IoT core rules.

### 2.4. Amazon Alexa

Amazon Alexa skills employ ASR and natural language understanding technology [60] for the development of custom applications that take vocal commands as input, convert them into text, interpret them to execute various actions, and possibly return a vocal response to the user that invokes the application. Alexa skills can be configured to send messages to various endpoints, including Lambda functions, essentially turning vocal commands into Lambda triggers.

The components of an Alexa skill invocation are the following:A wake word, such as “Alexa”, to turn on the device;An invocation name, the de facto name of the Alexa skill that is being invoked;An utterance, telling the Alexa skill what action to execute, or, in Alexa skill jargon, which “intent” is to be called;Optionally, a series of custom “slots”, which are additional variables that are being passed to the intent.

Intents produce a message in JSON format that is then forwarded to a Lambda endpoint. The Lambda function will parse the JSON message content, call a dedicated handler based on the intent, execute the handler’s routine, and lastly produce a JSON response message that is sent back to the Alexa skill.

## 3. Materials and Methods

The design proposed in this paper is applicable to any kitchen or home appliance due to the flexibility of the MQTT protocol and the serverless paradigm. It provides a valuable framework for the management of smart appliances that can be easily replicated in any smart home system. It leverages technologies in the field of serverless and machine-to-machine communication to create interactive smart objects with dedicated on-demand services, with good accessibility due to the use of vocal interfaces.

A description of the proposed framework for a smart appliance system follows in this section, and in particular, the design of the hardware and the serverless application for a smart kitchen extractor fan are outlined. The framework is completely application agnostic and can be replicated with any kitchen appliance or smart home device. The cloud-developed application makes use of Amazon Alexa to build a vocal interface and provides touchless interaction with the smart object. A phone application acting as an MQTT publisher can also be used to send commands to the appliance. MQTT makes the framework extremely flexible, and it makes the integration with cloud platforms (in this case, with AWS) simple and secure due to the use of certificates securely stored on the device by the use of a cryptochip. MQTT is also used to receive status updates from the smart object. The serverless framework provides efficient replicability and scalability properties to the application, and most importantly, it rationalizes resource allocation on the cloud when managing a large number of devices.

### 3.1. Smart Extractor Fan Design

The kitchen extractor fan is controlled by a device whose block diagram is illustrated in Figure 2. It consists of a microcontroller (referred to as “micro-host”) that interfaces with various peripherals and controls the ventilation motor and a set of lights. A NB-IoT chip is used to receive commands from the cloud in MQTT format over NB-IoT. The MQTT protocol guarantees a simple and secure interface toward the cloud through the native use of TLS encryption and authentication. The AWS IoT core requires authentication over MQTT with X.509 certificates. A cryptochip is used to securely connect and authenticate the AWS IoT core. Such a device can securely store certificates on the IoT device and authenticate the AWS IoT core.

The device receives messages from the cloud, parses them, and executes commands based on the content of their payload. It also sends state updates toward the cloud. Firmware updates can be received over the air using MQTT, simplifying the deployment of new firmware on large fleets of devices.

### 3.2. Architecture of the Serverless Cloud Application

An application was designed for the management and control of smart objects. The application is developed using AWS Lambda and the MQTT protocol to provide flexible interaction between the users and the extractor fan. Amazon Alexa is used to implement a vocal interface for the control of the smart object. State updates are also sent by the fan using the custom commands protocol and are processed by a dedicated Lambda function on the cloud to keep track of the state of each device.

The application was developed on AWS, and the serverless platform of choice was Amazon Lambda. Two Lambda functions were implemented in NodeJS (JavaScript) using the AWS SDK: one to forward user commands to devices, either by an application or Alexa skill, and the other to handle state updates from the devices. JavaScript, being a high-level interpreted language, lends itself quite naturally to the development of serverless or FaaS applications and, alongside Python, is one of the most used languages for the development of serverless applications [3]. Lambda functions comprised one or more handlers that are used to manage triggers, which can be received by various outside services, including Amazon Alexa or AWS IoT core. Each handler was defined on the basis of the custom command protocol and of the intents that were implemented in the Alexa skill. This way, users are able to control and monitor all the admissible states of the devices.

The project uses two AWS storage services: Amazon S3 and Amazon DynamoDB. DynamoDB is used to store devices states and devices and user identifiers. Similarly to the approach used in [61] to store Lambda code, in this project, Amazon S3 is used to store the device firmware. This way, each device can download any firmware version at any moment, greatly simplifying software updates and the deployment of new code. A software update can be initiated by sending the url of the S3 bucket containing the new firmware inside an MQTT command. The device will then download the file from the S3 bucket and execute the firmware update in autonomy.

An Alexa skill has been developed to interact with the devices. The endpoint of the skill is Amazon Lambda. A Lambda function is triggered when a command is invoked by the user. The Alexa skill takes care of translating input vocal commands into text and then translates text into JSON inputs that can be parsed by Lambda functions. Each JSON input produced by the Alexa skill also contains a user identifier that is used by Lambda to retrieve (from DynamoDB) the device ID associated with the user that invoked the skill. This mechanism guarantees that Lambda functions are only able to control the fans corresponding to the user that invoked the Alexa skill from their personal account. Custom slots are used to define a series of extra identifiers to distinguish between different fans when a user owns or controls more than a single fan. Figure 3 illustrates the message flow of the application. Green boxes indicate vocal commands and responses, yellow boxes are messages in JSON format generated by the Alexa skill, and the pink box indicates the payload of the MQTT message generated by the AWS application and forwarded to the device.

The complete description of the smart appliance system is illustrated in Figure 4. When a user invokes the Alexa skill with a valid intent, i.e., an Alexa skill intent for which a handler was defined, the skill triggers Lambda and passes a JSON message to it, containing both the UserID and a string identifying the handler required to manage that request. The Lambda function queries DynamoDB for the identifier(s) of the device(s) owned by that user. If the user owns more than one device, then a custom slot in the Alexa intent is used to distinguish between their multiple devices and retrieve the correct identifier from DynamoDB. Once Lambda knows the device to send the command to, and on the basis of the intent, it publishes an MQTT message on AWS IoT core following the commands protocol for that specific device. The device, having subscribed to its individual topics at start-up, will receive the message, parse its payload, and execute a command according to its content. Whenever it receives a command, the device will also publish an update on its status on its dedicated OutState topic. It is also possible to configure a “refresh time” at the monitoring topics, i.e., a time at which the devices sends periodic updates on the topic. An AWS IoT core rule is implemented to handle state updates and to invoke a second Lambda function, which updates a DynamoDB table with the new state information. The state information is used to monitor devices and their activities, but it also allows users to ask the Alexa skill for the state of the fan, and it uses “feedback” commands that require the Lambda function to know the current state of the device (e.g., “lower the ventilation”, “raise the ventilation”, and similar commands). The implementation of feedback intents makes the communication between the smart appliance and the user feel more “natural”. At the same time, periodic refreshes at short intervals amplify the application load on the cloud and the number of transmissions, which might not be suitable for certain devices or communication protocols. For such use cases, it might be required to set a longer refresh time or to only enable refreshes when there is a state update via outside command. The refresh option has been used for the experimental evaluation of the smart object work load on the cloud.

## 4. Experimental Evaluation

An experiment was carried out to evaluate the use of narrowband IoT to dispatch MQTT messages across MQTT clients. The goal of the experiment was to provide an estimation of the time needed to forward a message to/from an MQTT client over NB-IoT to show whether the protocol is suitable for applications, such as the one developed in this paper, that require an acceptable latency performance to ensure a good user experience.

The experiment involves the following nodes

An MQTT Client (Client 1) that publishes messages on a dedicated topic named Topic1;An MQTT broker;A NB-IoT module subscribed to Topic1 and publishing on a dedicated topic named Topic2;A second MQTT Client (Client 2) subscribed to Topic2.

This experimental setup is shown in Figure 5. The two MQTT clients might represent cloud applications sending or receiving messages from a smart object, such as vocal commands or state updates, while the NB-IoT device stands for the smart object itself. This setup reflects the interaction between users and the smart extractor fan system, where client 1 might be a Lambda function publishing a command on the broker to control the NB-IoT module/smart fan, which will then immediately publish a status update on the OutState topic once it sets the correct output variables.

At the start of the evaluation, the board registers to the NB-IoT network, connects to the MQTT broker, and it subscribes to Topic1. At the same time, the two clients also establish a connection to the MQTT broker, and Client 2 subscribes to its assigned topic. After waiting a short time to ensure that each MQTT client has successfully connected to the broker, Client 1 will publish a first message on Topic1, containing a timestamp T1 representing the start of the communication. The broker then forwards the message to the evaluation board. The board will then publish a new message on Topic2 containing T1 and the time elapsed between reception of the first message from the broker and the transmission of the second message. The broker will forward this message to Client 2, which will save the timestamp of reception (T2). This information, alongside the one obtained from the first client, the board, and the broker, is used to evaluate latency at different sections of the communication. The procedure is repeated multiple times at intervals of 10 s between each message.

The two MQTT clients were implemented on a computer using the Paho-MQTT [62] Python library. The MQTT broker was deployed on a different machine in the same network. The broker of choice was the Eclipse’ Mosquitto [63] broker. The MQTT clients are synchronized using the network time protocol. The NB-IoT module is connected to a mobile IoT network, which provides acceptable network coverage and reliability. In debug mode, Mosquitto does not provide a precision for packets logging below the second. However, the time difference between the transmission of the packet and its reception on the broker is not higher than a second, as verified on the Mosquitto log file. Therefore, considering symmetrical links and a negligible delay for the communication between the clients and the broker on the same network, the communication delay between the broker and the smart object was estimated as
$$T_{NB-IoT}=\frac{T_{2}-T_{1}-\Delta }{2},$$
in which Δ represents the time elapsing between the reception of the message on the board and the transmission of a message toward the broker.

## 5. Results

The performance of the developed smart fans was monitored using Amazon Cloudwatch and forcing a situation with frequent calls to Lambda. By setting a refresh time of 1 min for the OutState topic, the Lambda function responsible for updating the state database is called once every minute by the AWS IoT rule that handles messages received on the OutState topic. The function simply parses the JSON message forwarded to it by AWS IoT core, containing the new states and the identifier of the smart object, and updates a DynamoDB table with this new information, also increasing a usage atomic counter on the table to keep track of the usage of each smart device. Such a frequent refresh time might be incompatible with NB-IoT, given the high transmission load compared to the relatively low throughput expected of applications using this protocol, but more suitable to WiFi implementation. Furthermore, configuring the device so that the OutState topic is refreshed only when there is a state variable change is an implementation choice with an acceptable load for NB-IoT, and possibly a more realistic work condition for the smart home object. In the case of the Cloudwatch evaluation, the goal was to evaluate a situation with frequent calls to the serverless Lambda function; therefore, the number of state updates was incremented.

The results of the evaluation are shown in Figure 6 and Figure 7. The device was left on for about 24 h and the Cloudwatch Lambda insights service was used to keep track of various parameters. Figure 6 shows the median, 90th percentile, and maximum duration of Lambda function calls, computed in intervals of 5 min. Figure 7 shows the maximum CPU usage time and network load during 24 h, computed in 5-min intervals.

The spikes in duration correspond to “cold starts” that were also recorded by Cloudwatch. A cold start is the time required to allocate cloud resources to instantiate a Lambda function environment for the first time. Each Lambda function is only kept “warm” for a certain amount of time, requiring a new, prolonged startup to instantiate a new execution environment once the execution environment turns “cold”. Setting up provisioned concurrency can help mitigate the effect of cold starts when there are multiple devices invoking the Lambda function.

The results of the experimental evaluation carried out to evaluate the use of narrowband IoT to send MQTT messages through MQTT clients are shown in Figure 8. The delay has a mean value of 0.447 s, with occasional spikes of up to 11.41 s. The 90th percentile is 0.527 s, and the standard deviation is 0.938 s. The contribution of Δ is negligible and has minimal variations across multiple repetitions.

To investigate the reason behind the spikes, the experiment was repeated keeping track of packet numbers and MQTT acknowledgment (ACK) messages received by the board. Having set the MQTT quality of service to 1, the MQTT broker forwards an ACK message toward the board whenever a message has been published correctly. The NB-IoT board firmware was therefore slightly modified so that, after an ACK is received from the broker, a second MQTT message containing the current ACK number and Δ is also immediately published on Topic2. Client 2 therefore keeps track of the packet numbers, their associated timestamps, and the ACK numbers, besides computing the time difference between the timestamp of reception and the timestamp sent by Client 1, as in the previous experimental setup. The new experiment flow is detailed in Figure 9.

The iteration of the experiment described in the Figure comprises the following steps. At first, a packet P1 containing a timestamp T1 and packet number N1=1 is sent to the broker. P1 is then forwarded by the NB-IoT board and is correctly acknowledged; therefore, nACK1 is also set to 1 and sent to Client 2. The difference between T1 and the timestamp of reception (T2) is computed by Client 2 and is referred to as T21 in the figure. The delay in this case is below a reasonable value, i.e., not corresponding to a latency spike. During the following transmission, no ACK is received for packet P2, and thus, *n*ACK is not increased and P2 is reinserted in the output queue. At the third transmission, the board sends both P2 and the new P3 packet and receives an ACK for both messages. nACK for P3 is not set to 3 in this case to keep track of the previously lost ACK message, meaning that both P2 and P3 have an associated nACK=nACK2=2. By looking at log entries on Client 2 where a packet has an associated packet number that is different from its ACK number, it is therefore possible to find packets for which the previous ACK message was not received by the board.

These packets also correspond to spikes in latency in the resulting latency graph, meaning that the spikes are most likely due to the development of an output queue in the NB-IoT board. The loss of an ACK message could be because the publish message did not reach the broker or the ACK message from the broker was lost in transmission. In this last case, since MQTT QoS is set to 1, there is no guarantee that each message will not be forwarded twice toward Client 2. Client 2 does not receive any duplicate message, meaning that the reason for a lack of ACK messages on the NB-IoT board is most likely because the original publish message traveling on the NB-IoT network was lost. The average loss across the two experiments is about 0.8%. The latency for the second experiment was in line with the previous one, even if slightly lower (0.3845 s).

## 6. Conclusions

In this paper, a generalized and flexible framework for smart object applications in a smart home environment was proposed, leveraging MQTT, the serverless computing platform Amazon Lambda, Amazon ASK, and NB-IoT. The framework consists of smart objects receiving messages from the cloud through a dedicated message-exchange protocol and interactive vocal interfaces, with serverless functions deployed on the cloud to monitor and control the objects. A practical use case based on this framework was also developed, prototyped, and tested. The smart object, a smart kitchen fan, is equipped with a NB-IoT module and makes use of a custom command exchange protocol based on MQTT. A vocal interface was developed using Amazon Alexa and Lambda, enabling vocal control of all the smart object’s controllable states. The suitability of NB-IoT for MQTT messages dispatching was tested and showed a good latency performance, excluding some peaks that were evaluated and traced back to the formation of MQTT queues on the sender due to lost packets. Excluding cold starts, Lambda execution times are below 200 ms, positively contributing to an overall quasi-real time interaction between users and the smart object. Such a connected object could integrate additional customized services such as predictive maintenance and consumption control, enabled by cellular connectivity and simplified by resource provisioning on the cloud.

## Figures and Tables

**Figure 1 sensors-23-04459-f001:**
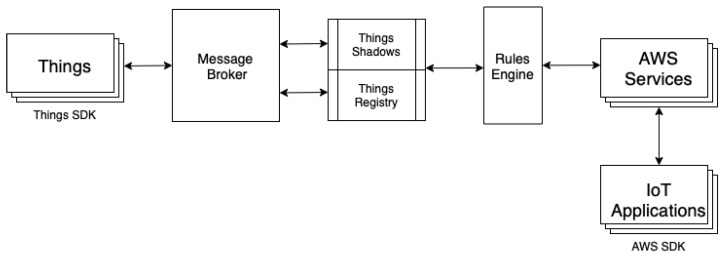
Architecture of AWS IoT core (adapted from [9]).

**Figure 2 sensors-23-04459-f002:**
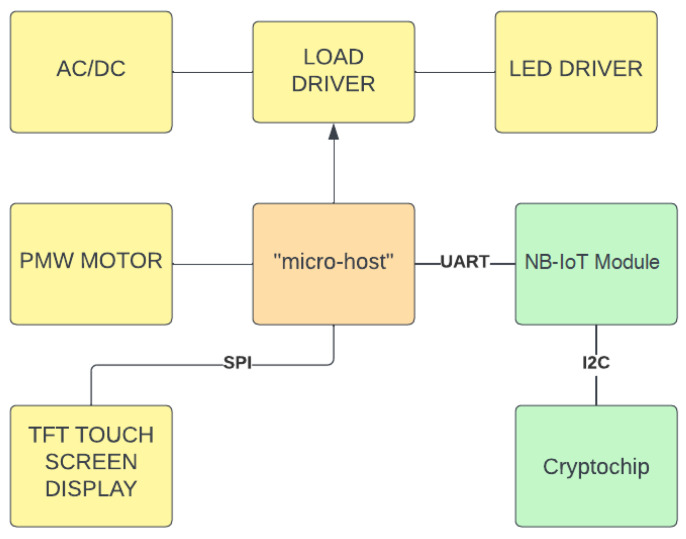
Block diagram of the smart extractor fan.

**Figure 3 sensors-23-04459-f003:**
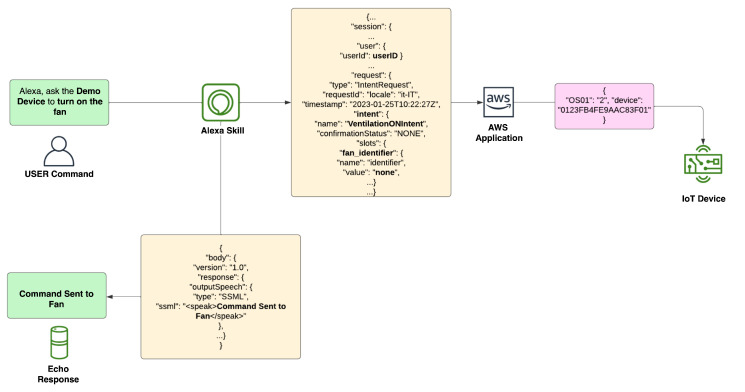
Messages flow from the user command up to the MQTT message generated by AWS that is forwarded to the device.

**Figure 4 sensors-23-04459-f004:**
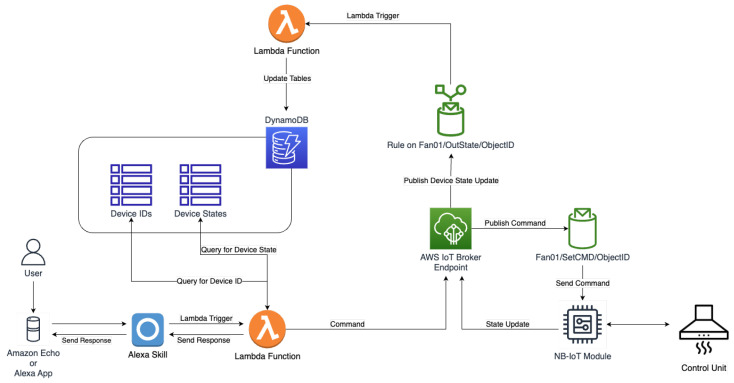
Architecture of the proposed smart fan application.

**Figure 5 sensors-23-04459-f005:**
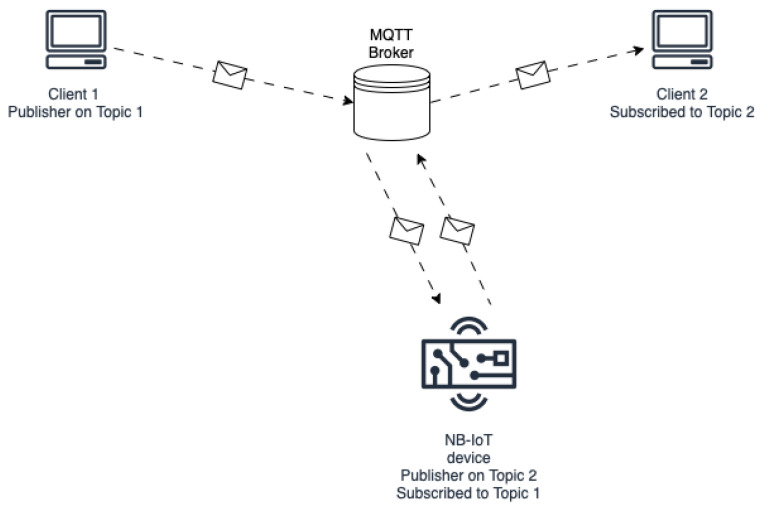
Diagram of the evaluation testbed used to evaluate NB-IoT latency for dispatching MQTT messages.

**Figure 6 sensors-23-04459-f006:**
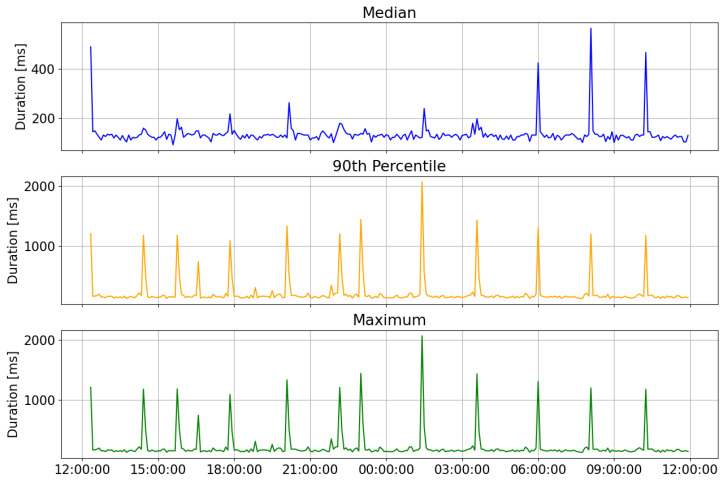
Median, 90th percentile, and maximum duration of a lambda function calls, computed in intervals of 5 min.

**Figure 7 sensors-23-04459-f007:**
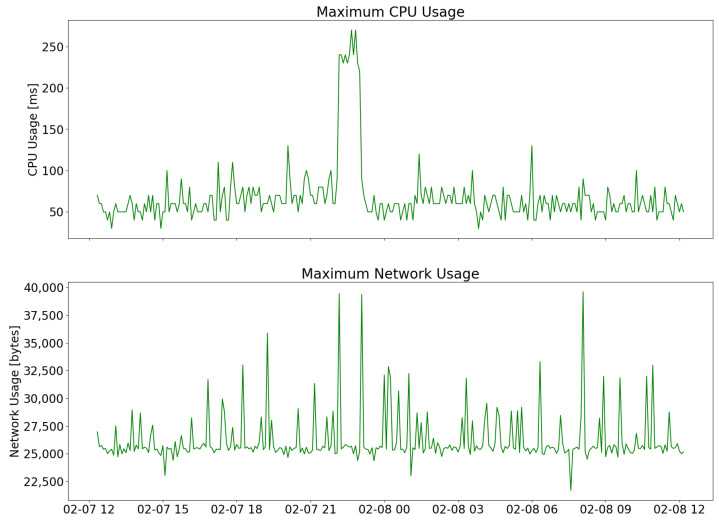
Maximum CPU (**top**) and network (**bottom**) usage during the Lambda function testing, computed in intervals of 5 min.

**Figure 8 sensors-23-04459-f008:**
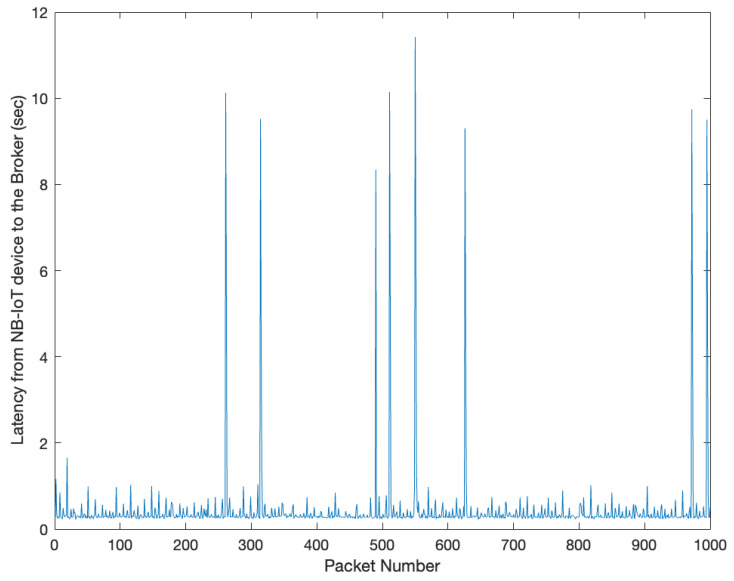
Estimated latency from the NB-IoT device to the MQTT broker.

**Figure 9 sensors-23-04459-f009:**
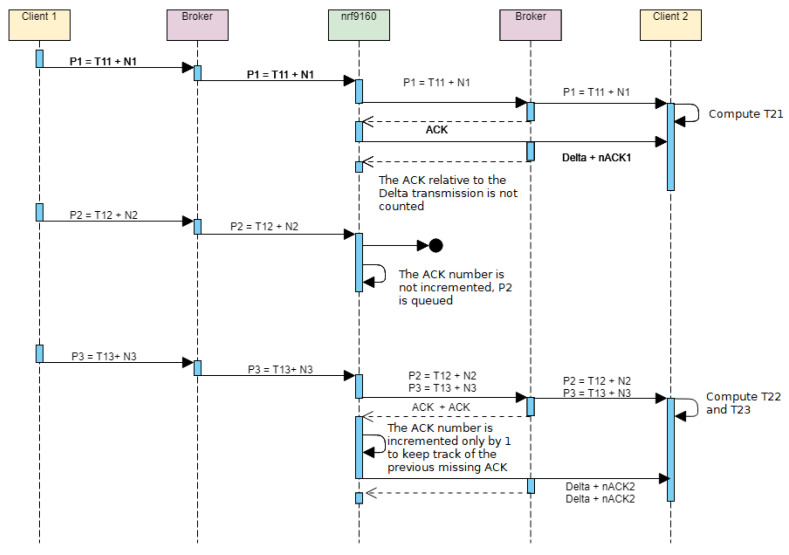
Sequence diagram of the experiment used to investigate latency spikes by keeping track of packet numbers and ACK messages.

**Table 1 sensors-23-04459-t001:** Comparison of communication protocols for IoT and machine-to-machine applications.

	NB-IoT	LoRaWAN	Sigfox	LTE-M
Spectrum	Cellular (LTE)	ISM	ISM	Cellular (LTE)
Data Rate [46,47]	∼200 kbps	∼0.3–100 kbps	∼100–600 bps	∼1 Mbps
Payload (bytes) [47,48]	1600	243	12	1000
Tx Power (dBm) [48]	23	13	14	23
Consumption [48]	Medium low	Low	Very Low	Medium

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
