# Peer review of "Design and Implementation of a Framework for Smart Home Automation Based on Cellular IoT, MQTT, and Serverless Functions"

_sensors, 2023, doi:10.3390/s23094459_

Round 1

Reviewer 1 Report

The paper proposes a framework for home automation applications that can be applied to control and monitor a device or object in a smart home environment.

The Title and Abstract clearly define the subject of the work.

The Introduction is very interesting and presents well some studies published in the literature. In the last paragraph of the Introduction, “section 4” of this article was missing.

Section 2 presents the main technologies implemented in this article, in addition to relating well the state of the art of each subject of interest. The quality of Figure 1 can be improved.

Figures 2, 4 and 5 need improvement. The resolution of the images is low and the text is very small.

In Figure 3, there is the impression that some blocks could have unidirectional or bidirectional arrows to better explain the interconnection between blocks.

Section 3 sufficiently and clearly describes the proposed framework implemented.

The authors chose to separate the experiments (section 4) from the results (section 5) into two sections, which facilitated understanding.

Figures 7 and 8 need to increase the text size.

Figure 10 could be improved by slightly increasing the text size.

The authors' conclusions are relevant to the objectives of the article and the results achieved.

Reviewer 2 Report

The article presents an analysis of a procedure for controlling a device in the home using a voice assistant and an LPWAN channel (NB_IoT); the validity of the topic is more extensive. From a technological point of view, it is not new; the article is an analysis of system integration.

The part of interest concerns the experimental results; these, given the nature of the protocols used, do not present any major surprises. 

From my point of view, the article is a long description of what is well-known and does not present any research results. The article is an evaluation of the performance of system integration. Still from my point of view, the work should be recalibrated: shortened, tested under different conditions (e.g. in conditions of low energy availability - the vacuum example might not be a good example-), and by comparing NB-IoT with other communication protocols aimed at comparable applications (e.g. Lora, WirelessHART, Sigfox, 5G).

(MDPI) Díaz Zayas, A.; Rivas Tocado, F.J.; Rodríguez, P. Evolution and Testing of NB-IoT Solutions. Appl. Sci. 2020, 10, 7903. https://doi.org/10.3390/app10217903.

Reviewer 3 Report

This paper presented a communication framework for smart object/things control and monitoring in a smart home. A message exchange protocol based on MQTT is proposed that communicates with cloud-deployed services. The researchers develop a practical use case where a home extractor fan equipped with an NB-IoT module is controlled
Via the Amazon cloud services. The framework is implemented and successfully deployed and tested.

The authors need to address the following comments to improve the paper:

1. The introduction can be shortened by removing verbose text and excessive details about Amazon services such as Alexa (ASK). A reference to relevant research works is sufficient. The authors merely report on recent works without connecting any of them to their contribution. Reading through the introduction, it is not clearly stated what is the research problem the authors are addressing and what is their contribution.

2. Theoretical background section also simply reports on the NB-IoT, MQTT, and Amazon services listing citations without providing any details on recent work. It is important to see how authors' contributions contrast with the existing work and how it advances the knowledge in the area.

3. Figure 1 should cite a reference where it is taken from.

4. Figure 2 is simply replicating what is presented in figure 5. It can be removed or merged with fig 5.

5. Page 6: line 247:  It is stated that MQTT makes the integration with “Cloud Platforms (in this case, with AWS) simple and secure.” Is there a reference for this? I do not see that this work is focused on security.

6. Page 8: line 304: “When a user invokes the Alexa Skill with a valid intent,” please elaborate what is “valid intent”.

7. Page 9: line 314: what is “refresh time,” and How often is it triggered? Would it not negatively affect the overall performance of the framework, as suggested in line 323?

8. Page 10: line 368-369: Please rephrase “However, there is no difference higher than a second between T1 (time of transmission of the first packet) and the corresponding timestamp on the Mosquitto log file.”

9. Page 10: lines 366 - 382: It seems the text is duplicated including the equations. Both equations are identical.

10. Page 13: line 435: What is “reasonable value”?

11. Page 13: line 445; what is the size of the output queue? In the subsequent lines loss of packets and multi-packet forwarding is listed. What was the latency and error rate for this experiment?

12. It is quite confusing to see keywords framework, architecture, and protocol(s) being used (sometimes) interchangeably throughout this manuscript.

Round 2

Reviewer 3 Report

The manuscript has improved, however I have these two comments that should help further improve it.

1. At the end of section II. Background, I would like to see how the authors fuse the technologies together to come up with their contribution. My original comment was not addressed.   2. Section III can include more details about relevant works. I'm sure the authors can find interesting works in these papers, to expand this section. A suggestion is to merge sections II and III into one section.   Krishnamurthi, R.; Kumar, A.; Gopinathan, D.; Nayyar, A.; Qureshi, B. An Overview of IoT Sensor Data Processing, Fusion, and Analysis Techniques. Sensors 202020, 6076. https://doi.org/10.3390/s20216076   B. Mishra and A. Kertesz, "The Use of MQTT in M2M and IoT Systems: A Survey," in IEEE Access, vol. 8, pp. 201071-201086, 2020, doi: 10.1109/ACCESS.2020.3035849.  

Tariq, M.A.; Khan, M.; Raza Khan, M.T.; Kim, D. Enhancements and Challenges in CoAP—A Survey. Sensors 202020, 6391. https://doi.org/10.3390/s20216391

The methods and results section has improved.
